# Monitoring Solar Radiation UV Exposure in the Comoros

**DOI:** 10.3390/ijerph181910475

**Published:** 2021-10-05

**Authors:** Kévin Lamy, Marion Ranaivombola, Hassan Bencherif, Thierry Portafaix, Mohamed Abdoulwahab Toihir, Kaisa Lakkala, Antti Arola, Jukka Kujanpää, Mikko R. A. Pitkänen, Jean-Maurice Cadet

**Affiliations:** 1LACy, Laboratoire de l’Atmosphère et des Cyclones, UMR 8105 CNRS, Université de La Réunion, Météo-France, 97744 Saint-Denis, France; marion.ranaivombola@univ-reunion.fr (M.R.); hassan.bencherif@univ-reunion.fr (H.B.); thierry.portafaix@univ-reunion.fr (T.P.); jeanmaurice.cadet@gmail.com (J.-M.C.); 2School of Chemistry and Physics, University of KwaZulu-Natal, Durban 4041, South Africa; 3Agence Nationale de l’Aviation Civile et de la Météorologie, Moroni 84646, Comoros; fahardinetoihr@gmail.com; 4Space and Earth Observation Centre, Finnish Meteorological Institute, 99600 Sodankylä, Finland; kaisa.lakkala@fmi.fi (K.L.); Jukka.Kujanpaa@fmi.fi (J.K.); Mikko.Pitkanen@fmi.fi (M.R.A.P.); 5Climate Research Programme, Finnish Meteorological Institute, 70211 Kuopio, Finland; antti.arola@fmi.fi

**Keywords:** ultraviolet radiation, clouds, tropics, observations, modeling, erythemal doses

## Abstract

As part of the UV-Indien project, a station for measuring ultraviolet radiation and the cloud fraction was installed in December 2019 in Moroni, the capital of the Comoros, situated on the west coast of the island of Ngazidja. A ground measurement campaign was also carried out on 12 January 2020 during the ascent of Mount Karthala, located in the center of the island of Ngazidja. In addition, satellite estimates (Ozone Monitoring Instrument and TROPOspheric Monitoring Instrument) and model outputs (Copernicus Atmospheric Monitoring Service and Tropospheric Ultraviolet Model) were combined for this same region. On the one hand, these different measurements and estimates make it possible to quantify, evaluate, and monitor the health risk linked to exposure to ultraviolet radiation in this region, and, on the other, they help to understand how cloud cover influences the variability of UV-radiation on the ground. The measurements of the Ozone Monitoring Instrument onboard the EOS-AURA satellite, being the longest timeseries of ultraviolet measurements available in this region, make it possible to quantify the meteorological conditions in Moroni and to show that more than 80% of the ultraviolet indices are classified as high and that 60% of these are classified as extreme. The cloud cover measured in Moroni by an All Sky Camera was used to distinguish between the cases of UV index measurements taken under clear or cloudy sky conditions. The ground-based measurements thus made it possible to describe the variability of the diurnal cycle of the UV index and the influence of cloud cover on this parameter. They also permitted the satellite measurements and the results of the simulations to be validated. In clear sky conditions, a relative difference of between 6 and 11% was obtained between satellite or model estimates and ground measurements. The ultraviolet index measurement campaign on Mount Karthala showed maximum one-minute standard erythemal doses at 0.3 SED and very high daily cumulative erythemal doses at more than 80 SED. These very high levels are also observed throughout the year and all skin phototypes can exceed the daily erythemal dose threshold at more than 20 SED.

## 1. Introduction

The ultraviolet radiation (UVR) coming from the sun and reaching Earth’s surface has a direct effect on human health, terrestrial and aquatic ecosystems, and the degradation of materials [1]. The effects of UVR on human health can be harmful. Overexposure to UV-B can cause DNA damage and increases the risk of skin cancers. The exposure to UVR remains one of the main causes of melanomas [2]. Behavior is an important factor, and chronic exposure was found to be a pathway to keratinocyte cancer [3]. Eyes are also affected by UVR. In fact, studies showed that UVR is the main environmental factor in the development of pterygium [4] and cataract [5]. Studies also found that high exposure to UVR can induce immunosuppression [6,7]. These effects are modulated by behavior and reasonable exposure to UVR can also have beneficial effects on human health. For example, the synthesis of vitamin D is activated by skin exposure to UV-B [8]. Recent studies showed that low exposure to UVR can also stimulate the immune system [9]. To increase public awareness and standardize the measurement of UVR harmful to human health, the Ultraviolet Index (UVI) was defined by the World Health Organization (WHO) and classified in an exposure scale ranging from “low” to “extreme”. Above 3 UVI units, long exposure can be harmful and protection is necessary. The 8–10 UVI threshold is defined as “very high” exposure, while the threshold above 11 UVI units is defined as “extreme”.

The UVI in the southern Indian Ocean was found to be higher than 10 UVI units almost throughout the year [10]. Extreme UVI, as high as 15 to 20 UVI units, was also measured at various location in this region such as Antananarivo (Madagascar), Mahé (Seychelles), Plaine Corail (Rodrigues), and Saint–Denis (La Réunion) [11,12,13]. This may imply dangerous consequences for the health of the local population. In La Réunion, a significant increase in the occurrence of skin lesions due to sun exposure was observed by dermatologists, and concerns about 2% of the population [14]. The surface values of UVI depend on multiple parameters, of which solar zenith angle (SZA), altitude, and latitude are the most determining. However, cloud cover may strongly affect surface UVI values. It can reduce UVI by as much as 80% [15] or increase it by about 20% [16,17]. In the southern Indian Ocean, evidence of frequent occurrence of UVI enhancement by clouds was found [11]. Depending on quantities and their nature, aerosols may also contribute to UVR variability at the surface by scattering or by absorption [18]. Surface albedo also modulates surface UVI values, and these modulations can be an increase to the order of a 40% for snow covered surfaces [19]. When UVR passes through the atmosphere, it also interacts with certain gases. Stratospheric ozone plays an important role in ground-based UVI values because of its strong absorption in the UV band. Ozone is produced in the tropical region and is advected towards the high latitudes by the Brewer–Dobson Circulation (BDC), a meridional large-scale circulation that is characterized by upwelling of tropical air masses, with poleward flow in the stratosphere, and downwelling near polar regions [20]. The concentrations of ozone were carefully studied and monitored. Significant depletion of ozone was observed over the last 40 years due to the emissions of ozone depleting substances (ODS) into the atmosphere. The Montreal Protocol (1987) and its amendments were successful in reducing the emissions of ODS, which are now close to historical levels [21]. Nonetheless, the increase of greenhouse gases (GHG) emissions in the atmosphere was suspected to accelerate the BDC [22,23]. Ozone would then be transported more rapidly from the tropics towards high latitudes, thus leading to less stratospheric ozone and increased of surface UVR in the tropics [24].

To study and monitor the evolution of surface UVR at various points in the southern tropics, an observation network was developed in the south-western basin of the Indian Ocean by the University of La Réunion. It is known as the UV-Indien network and is funded by Europe, the Reunion Council and CNES (the French Space Agency). The main objective of this network is to understand and anticipate the variability of surface UVR to alert populations to the UV risk, in a region where levels are extreme and means of measurement almost nonexistent. Among the 10 stations in the western tropical Indian Ocean, one station was set up in the Union of the Comoros. This small archipelago of 4 islands is located in the north and center of the Mozambique Channel. The UV-Indien measuring station is implanted in the meteorological center of the capital Moroni on the island of Ngazidja (also known as Grand Comoros, 11.708° S, 43.238° E). The Union of Comoros is an agricultural country with a population of about 870,000. It is classified as one of the least developed countries due to its low GDP per capita (about US $1371 ), high economic vulnerability, and low health indicators [25]. Total expenditure on health is 6% of the national GDP, and per capita this expenditure is around $101 [26]. Although there is a rural exodus, most of population is currently living in rural areas (72%) [27]. The country’s agricultural nature, the proportion of the population living in rural areas, and the low level of mechanization of agricultural work results in high exposure of outdoor workers to the sun. In addition, the island of Grand Comoros is formed by an active volcano called Karthala, and most of the island has significant relief. Much of the cultivated and inhabited land is located at altitude, which further increases the levels of solar radiation received by the population. However, as in most developing countries, health data in Comoros suffer from their discontinuity and low density. This does not allow the populations to be monitored or prevention strategies to be deployed, despite the risk of exposure to UV radiation associated with the very high to extreme UV indices during the whole year, whatever the season. These indices are largely due to the country’s tropical location, where the incoming solar radiation is very high and the total ozone column (TOC) has lower values than those at mid-latitudes.

## 2. Data and Methods

### 2.1. Data

#### 2.1.1. Ground Measurements

Field measurements of the UVI were conducted between the village of Mvouni, located at 390 m altitude, and the summit of Mount Karthala at 2360 m altitude on 12 January 2020. The objective was to measure UV indices and cumulative doses over the course of a day at regular time intervals, i.e., every 10 min. The ascent to the summit and the measurements started at 06:00 LT, and the return to the starting point took place at 17:00 LT. UVI measurements at the summit of the Karthala volcano were made at approximately 12:00 LT. For the purpose of this study, a Solarmeter 6.5 radiometer was used. This is a portable instrument containing a UV-sensitive photodiode [28] which allows the UVI to be estimated with an error of 10% (NIST traceable accuracy). The same Solarmeter 6.5 instrument was successfully used in previous works and similar situations [10,12]. Furthermore, as reported by [13], the Solarmeter instrument was regularly calibrated during side-by-side intercomparison campaigns with the Bentham DTMc spectrophotometer and shows a small discrepancy, with an overestimation of about 12%. For each UVI measurement with the Solarmeter, environmental observations were also made. These observations included sky conditions, following the WMO guidelines for sky observations (http://worldweather.wmo.int/oktas, accessed on 1 July 2021), and the extent of the solarmeter’s field of view: whether there was a clear hemispheric view or partial obscuration. UVI measurements and environmental observations were made approximately every 10 min.

In addition to the UVI measurements during the Karthala hike, continuous surface UVI measurements were available at the ANACM (Agence Nationale de l’Aviation Civile et de la Météorologie) site in Moroni (Figure 1). The ANACM site is part of the UV-Indien network [11]. It is equipped with a Kipp & Zonen SUV-E Radiometer, which was operating since December 2019. The instrument is located at 11.708° S, 43.247° E at 12 m above sea level. Measurements are taken every minute. The postprocessing includes a cosine correction and a correction based on the corresponding daily total ozone column and SZA. For the TOC, the OMTO3d product is used (Ozone Monitoring Instrument (OMI)/Aura Ozone Total Column Daily L3 Global 1 × 1 deg.). The last calibration of this instrument was carried out in July 2019. These UVI measurements will be called UVI-RADIO-AS hereafter. On the same site, a wide-angle total-sky imager was operating since December 2019. The camera is a SkyCamVision, manufactured by Reuniwatts (http://www.reuniwatt.com/, accessed on 1 July 2021). It acquires hemispherical images between 380 and 440 nm. Following a specific algorithm [29,30,31], a cloud fraction (CF) can be derived from these pictures. In addition to UVI and sky observations, colocalized meteorological datasets compiled by the ANACM were used to derive climatological values for temperature and rainfall, for the Moroni site and its surroundings. Quality control of the UV-Indien network data was carried out upstream during the preprocessing and calibration of the data as described in the article by [11].

#### 2.1.2. Satellite Measurements

Satellite UVI products were also used in this study. We retrieved satellite UVI from the OMUVBG product derived from the OMI on-board the Aura satellite. The OMI surface UV algorithm inherits from the surface UV algorithm developed by NASA for the Total Ozone Mapping Spectrometer (TOMS) and were further developed [32,33,34]. Daily UVI are computed with the OMI measurement, climatological ozone and temperature profile, albedo, elevation, and SZA. A clear-sky UVI is obtained, called UVI-OMI-CS hereafter. It is then corrected by a cloud modification factor based on the measured reflectance at 360 nm to obtain an all-sky conditions UVI, called UVI-OMI-AS hereafter. UVI-OMI-CS is calculated at the time of local solar noon, while UVI-OMI-AS is calculated at the time of local solar noon and at the time of the satellite overpass, i.e., at the exact time of the satellite measurement. The effects of absorbing aerosols are also taken into account following [34] and using the monthly climatology from [35]. The OMI pixel size at nadir is 13 × 24 km^2^.

The UVI product derived from the measurements of the TROPOMI instrument onboard the Sentinel-5P satellite were obtained from the Sodankylä National Satellite Data Centre (https://nsdc.fmi.fi, accessed on 1 July 2021). The TROPOMI UV algorithm [36] makes use of two precomputed lookup tables. Cloud optical depth is obtained from the first look-up table using the measured reflectance at 354 nm. Cloud optical depth, TOC from the TROPOMI L2 product [37], surface pressure, surface albedo and SZA are used to compute UVI at overpass time, and solar noon time for clear-sky conditions. The effects of absorbing aerosols are also taken into account using the monthly climatology from [35] following [34]. The resulting UV index used in this study will be called UVI-TROPOMI hereafter. From the TROPOMI product at the overpass of the Moroni station, UVI-TROPOMI located within less than 10 km from the ground station were selected. The TROPOMI pixel size is 5.6 × 3.5 km^2^ at nadir. The overpass files for the UV product version 1.02.02 (overpass extractor version 1.02.00) were downloaded from the FTP site accessible via the product web page (https://nsdc.fmi.fi/data/data_s5puv.php, accessed on 1 July 2021).

The Copernicus Atmosphere Monitoring Service (CAMS) provides UV irradiance forecasts in clear-sky and all-sky conditions (called UVI-CAMS hereafter). The UVI values from UVI-CAMS for the closest grid-point to the Moroni station were gathered for the period of this study. For each forecasted day, the first forecast was used to retrieve UVI-CAMS. The first forecast, which was initialized at 00:00:00 UTC, and provided output at timesteps 03:00:00, 06:00:00, and 09:00:00, gave UVI at the same time. The same process was implemented for the second forecast which was initialized at 12:00:00 UTC. Since 21 June 2016 the CAMS model has a horizontal resolution of approximately 40 km. As several improvements were made to the CAMS model, UVI-CAMS cannot be considered as a homogeneous time series. More information on CAMS models is available at https://atmosphere.copernicus.eu/node/326, accessed on 1 July 2021. Measurements from UVI-RADIO made in clear-sky conditions are compared against UVI-CAMS estimates for clear-sky conditions below, and the same logic applies for all-sky conditions.

These UVI satellite products were previously compared to measurements from four ground-based stations of the UV-Indien network located at St–Denis (Réunion), Mahé (Seychelles), Anse Quitor (Rodrigues) and Antananarivo (Madagascar) [11]. UVI-TROPOMI was found to underestimate ground-based measurements of UVI, with mean absolute differences of UVI units ranging between −0.43 ± 0.63 for St-Denis, −1.66 ± 0.62 for Antananarivo, −4.9 ± 0.5 for Mahé, and −1.97 ± 1.10 for Anse Quitor. For UVI-OMI, mean absolute differences of UVI units ranged between −1.03 ± 2.05, −0.33 ± 1.14, −3.2 ± 1.01, and −0.47 ± 2.28 at St-Denis, Antananarivo, Mahé and Anse Quitor, respectively. These comparisons were obtained in clear-sky conditions. In all-sky conditions, higher differences were observed and could be explained by insufficient satellite resolution of the sky conditions over the ground-based instrument, cloud enhancement, or by absorbing aerosol parameters different from those of climatology.

The characteristics of the different ground-based and satellites measurements used in this study are summarized in Table 1.

### 2.2. Methods

#### 2.2.1. Modeling

Radiative transfer modeling was also performed to compute surface UVI at different altitudes for the atmospheric conditions on 12 January 2020. We used the Tropospheric Ultraviolet Model (TUV) [43] to compute UVI based on the configuration described by Lamy et al. [44], which produced the best results in comparison with that of high-resolution ground-based UVI measurements made at St–Denis, Réunion Island. The station in Saint–Denis de la Réunion shares similar characteristics with the station in Moroni, Ngadzidja. Both stations are located on the side of an island in the southwestern basin of the Indian Ocean. Saint–Denis, is however, slightly further east and south in the Indian Ocean basin, at 20.902° S and 55.485° E. Under clear sky conditions, the UVI modeling had a bias of 0.5% at best with respect to ground-based measurements made with a Bentham DTMc300 spectroradiometer. This TUV configuration was adopted here, and solar spectral irradiance simulated at the Earth’s surface ranged from 280 to 450 nm with a 1-nm step. The vertical grid extended from the surface up to 80 km with 0.1 km resolution. The time resolution was 1 min. For the extra-terrestrial spectrum, we used Dobber et al. [45] spectrum and, for the ozone cross-sections, we took those given by Gorshelev et al. [46] and Serdyuchenko et al. [47]. Unfortunately, no ground-based measurements of TOC or aerosols are available in Moroni. The TOC used was obtained from the OMTO3d product, and aerosols and albedo were obtained from the level 3 daily MODIS product (MOD08), which includes aerosol optical depth, single scattering albedo and angstrom exponent. The monthly climatlology of vertical profile of ozone and temperature for multiple latitudinal bands obtained by McPeters and Labow [48] were used. The UVI values were modeled every minute for every 100 meters of altitude. They will be referred as UVI-TUV-H hereafter. The UVI values at sea level, called UVI-TUV-M hereafter, were also extracted.

#### 2.2.2. Clear-Sky Filtering

Two filtering methods were used to distinguish UVI measurements made during clear-sky and cloudy conditions, called respectively UVI-RADIO-CS and UVI-RADIO-AS hereafter. The first filtering method was carried out manually, each daily UVI profile was inspected and compared against an estimate of the clear-sky UVI based on Madronich’s analytical formula [49] and TOC of the TROPOMI instrument. A daily clear-sky UVI profile typically has a bell shaped curve. Thus, by comparing the clear sky analytical profile with the measured daily profile, a visual check was done for each day to deduce the periods of the day-time when the measured UVI deviated from the UVI bell-shaped profile due to attenuation by cloud. Periods of one hour clear sky data were then selected following this process. Finally, we obtained a filtered clear-sky UVI, called UVI-RADIO-CS-MF hereafter.

For the second method, a threshold value of the CF obtained from the All-Sky-Camera (ASC) measurements was used to filter the UVI-RADIO-AS. Values of UVI-RADIO were kept only when the CF was below a certain threshold and when the SZA was below 45°. In addition, the UVI-RADIO values were not taken into account when there were no CF values. This gave the automatically filtered UVI-RADIO clear sky called UVI-RADIO-CS-AF hereafter. Several CF threshold values were tested. To obtain the appropriate threshold value, the result of the first filtering method was compared with the second filtering method. The threshold value of the CF at 0.3 provided the closest possible filtering to the first manual filtering method. An agreement of 97.01% was obtained between the two methods.

## 3. Results

### 3.1. Climatological Description of UVI, Temperature and Rainfall at Comoros

Figure 1 shows the monthly climatological values of temperature and rainfall computed from the meteorological data (right panel). We can observe a seasonal variability driven by the annual oscillation that defines the wet season (from November to April) and the dry season (from May to October). During the wet season, due to the influence of the sea-breeze, convective clouds appear at mid-height on the east coast late in the morning and then overflow to the west of the coast in the afternoon. Monthly climatological values of All-Sky UVI, at overpass time (called UVI-OMI-AS-OP hereafter) and at solar local noon (called UVI-OMI-AS-LN hereafter), and Clear-Sky UVI at local noon (called UVI-OMI-CS-LN hereafter) estimated by OMI were computed over the period 2005–2020. The following variables are represented on Figure 2: the mean UVI-OMI-AS-OP (blue line) with the standard deviation (blue shading) on the left; the mean UVI-OMI-AS-LN (purple line) with the standard deviation (purple shading) in the center, and the mean UVI-OMI-CS-LN (yellow line) with the standard deviation (yellow shading) on the left. For these three variables, the median is represented by the black dot, the two edges of the plot correspond to the 25th and 75th percentiles, and the whiskers show the extreme values, excluding outliers (single black dots). Data points are considered outliers if they deviate by about ±2.7 
σ
. UVI-OMI-CS-LN ranges from 6 to 16 and, as expected, is slightly higher than UVI-OMI-AS-LN by about 1 point of UVI depending on the month. The difference in UVI is related to cloud attenuation because, unlike UVI-OMI-CS-LN, UVI-OMI-AS-LN takes the effects of cloud cover into account. UVI-OMI-AS-LN from 6 to 15, standard is larger, especially during the months of December, January, February, and March. These months correspond to the rainy season (Figure 1). UVI-OMI-AS-LN is higher than UVI-OMI-AS-OP, this is due to the time of the overpass which is offset by about 1 hour later on average from the local solar noon, and therefore the SZA associated with the overpass is higher. Thus, a higher SZA will result in a lower UVI. The annual relative frequencies of UVI-OMI-AS-LN are plotted in Figure 3 for the 2005–2020 period. The color code corresponds to that defined by the WHO, with low UVI between 0 and 3, moderate UVI between 3 and 6, high UVI between 6 and 9, very high UVI between 9 and 11 and extreme UVI above 11. Throughout the year, about 50–60% of the UVIs are considered extreme, and about 20–40% of the UVIs are classified as strong to very strong. The UVI-OMI-AS-LN measurements were recorded at local solar noon, and therefore these high to extreme UVI values are generally close to the observed daily UVI maxima. They thus underline the very high health risk run by the Comorian population almost daily due to exposure to UV radiation.

The average diurnal cycles of UVI and CF are plotted in Figure 4. The upper plot (Figure 4a) shows the total CF (blue downward triangle), the thick cloud CF (blue leftward triangle), and the thin cloud CF (blue rightward triangle). The middle plot (Figure 4b) shows the mean diurnal cycle of UVI for UVI-AS (purple circles), UVI-TUV-CS (blue triangles), UVI-RADIO-CS-MF (orange crosses) and UVI-RADIO-CS-AF (red squares). The lower plot (Figure 4c) depicts the number of UVI measurements for each minute of the day for UVI-RADIO-AS (purple line), UVI-RADIO-CS-MF (orange line) and UVI-RADIO-CS-AF (red line). UVI-TUV-CS-SUBSET (green triangle) is a subset of UVI-TUV-CS. Only the modeled points that correspond to a clear sky measurement (with UVI-RADIO-CS [AF] as a reference) at the same date and time form the subset. The diurnal cloud cycle shows a CF total of about 0.55 at the beginning of the day, a slight decrease is observed in the morning, followed by an increase for the second part of the day until a maximum CF of about 0.8 is found around 17:00 local time. The maximum cloudiness observed corresponds to the maximum cloudiness by thick clouds. The variability of the thick clouds dominates the variability of the total cloudiness, as the thin clouds appear early in the morning, around 8:00 local time, and are constant throughout the day with a CF of about 0.15. The average diurnal cycle of UVI shows a maximum at about 12 for UVI-TUV-CS, about 11 for UVI-RADIO-CS-MF and UVI-RADIO-CS-AF, and about 8 for UVI-RADIO-AS. UVI-TUV-CS shows no irregularity in the shape of the curve, the modeling was carried out under clear sky conditions only, the highest UVI profile and a well-defined bell-shaped curve were expected. UVI-TUV-CS is not the absolute reference but an attempt to obtain an average clear-sky profile. This model is dependent on the uncertainty related to the input parameters of the radiative transfert model, such as the TOC or parameters of aerosols (optical thickness, angstrom coefficient and single scattering albedo) and on the uncertainty of the radiative transfer scheme used, the latter being about 5% [50]. We see in the next section (Section 3.2) that a significant bias is obtained between the ground-based clear-sky measurements and the clear-sky modeling via TUV. UVI-RADIO-AS contains all types of sky, cloudy or not, which explains the lower UVI values throughout the day. In the case of UVI-RADIO-CS-MF and UVI-RADIO-CS-AF, we expected to find at least a well-defined bell curve and maximum UVI values of the same order as UVI-TUV, which is not the case here. This is due to the small number of points per minute of these two data sets, between 12:00 and 18:00 local time, UVI-RADIO-CS-MF and UVI-RADIO-CS-AF have only 40 to 80 points per minute (Figure 3c). This strongly reduces the statistical significance. The period studied is still too short to obtain enough clear-sky points per minute. The last panel shows the large difference between the number of measurements/minutes available in all sky conditions and in clear sky. The latter value is well correlated with the average diurnal evolution of cloud cover. In addition, the presence of clouds is more pronounced during the austral summer rainy season (Figure 1), and it is also during this season that clear-sky UVI is highest (Figure 2). Therefore, more measurements of high UVI are filtered out during this season than of moderately high UVI during the southern winter. This explains the values and shape of the mean diurnal UVI cycle for UVI-RADIO-CS-MF and UVI-RADIO-CS-AF. While UVI-TUV-CS covers the entire study period and has the shape of a Gaussian. UVI-TUV-CS-SUBSET has the same shape as UVI-RADIO-CS-MF and UVI-RADIO-CS-AF but is slightly higher than the latter two. The correspondence between these three curves reinforces the idea that their shape (not perfectly Gaussian at the top) is influenced by the small number of points available at that time of the day, which is insufficient to represent the diurnal cycle of UVI over the whole period.

### 3.2. Intercomparison between UVI Satellite Estimates or UVI Modeling Estimates and UVI Ground Measurements

UVI-RADIO-CS-AF and UVI-RADIO-AS measurements were compared with that of UVI satellite estimates and UVI model estimates from 12 December 2019 to 1 February 2021. To compare these measurements, a selection was made on the satellite or model estimates. UVI satellite estimates or UVI model estimates had to be located at less than 10 km from the Moroni measurements, and the time difference between the two measurements had to be no greater than one minute. If several points were selected, the average of these points was compared to UVI-RADIO-CS-AF and UVI-RADIO-AS. Figure 5 shows the correlation between ground-based measurements made with the radiometer and satellite or model estimates. All-Sky measurements are represented by blue dots while Clear-Sky measurements are represented by yellow dots. The three lines in the figure are: the identity line (dark line) and linear regression under All-sky and Clear-Sky conditions (red dashed line and red solid line respectively). Clear-sky satellites or model estimates of UVI (UVI-TROPOMI-CS, UVI-CAMS-CS, UVI-TUV-CS-SUBSET) were compared against ground-based clear-sky UVI (UVI-RADIO-CS-AF), and all-sky satellites or model estimates of UVI (UVI-TROPOMI-AS, UVI-CAMS-AS) were compared against ground-based all-sky UVI (UVI-RADIO-AS). The correlation coefficient, and relative and absolute differences, including their means and medians, were also calculated and are shown in Table 2.

Satellite and model estimates correlate better with UVI-RADIO measurements under clear sky conditions than under cloud sky conditions, with a correlation coefficient greater than 0.90 (0.93 for UVI-TROPOMI-CS, 0.98 for UVI-CAMS-CS, and 0.95 for UVI-TUV-CS-SUBSET). Under all-sky conditions, UVI-TROPOMI-AS and UVI-CAMS-AS show correlation coefficients of 0.68 and 0.55, respectively. Under all-sky conditions, between UVI-TROPOMI-AS and UVI-RADIO-AS the mean absolute difference in UVI is between 1.97 ± 1.69, and the median absolute difference is 1.57 UVI units. Between UVI-CAMS-AS and UVI-RADIO-AS, the mean absolute difference in UVI units is 2.45 ± 2.40 and the relative median is 1.66. The standard deviation is very high in both cases. These differences and deviations are explained by the difficulty satellites or models encounter in representing a UVI measured on the ground: the spatial resolution of the satellite products or models is relatively large (13 × 24 km^2^ for OMI and 5.6 × 3.5 km^2^ for TROPOMI) and is not fully representative of the sky condition just above the UV radiometer. The latter is strongly dependent on the cloud cover just above the instrument. This cloud cover will tend to attenuate the UVI in most cases, but increase the UVI in some cases [11]. Such attenuation or increase is not always taken into account by the satellites. A glance at the clear-sky comparison reveals that the mean and median differences in UVI between UVI-TROPOMI-CS and UVI-RADIO-CS and between UVI-CAMS-CS and UVI-RADIO-CS are much smaller: the median is 0.28 for the former, and 0.54 for the latter. The means and standard deviations are also lower; the mean absolute difference is 0.49 ± 0.74 between UVI-TROPOMI-CS and UVI-RADIO-CS, and it is 0.68 ± 0.58 between UVI-CAMS-CS and UVI-RADIO-CS. The number of comparison points is lower for TROPOMI, especially in clear sky, due to the small amount of clear sky data at the time of the TROPOMI overpass. CAMS is less affected by the small amount of clear sky data because CAMS produces 4 points per day. Finally, a mean absolute difference of 0.86 ± 0.75 can be observed between UVI-TUV-CS-SUBSET and UVI-RADIO-CS. It can be explained by the representativeness of the input parameters to the radiative transfer model since many parameters are derived from monthly climatologies or specific recommendations: the aerosols parameters were obtained from Kinne et al. [35], the vertical distribution of ozone and temperature were obtained from [48] and surface albedo were fixed following [51]. The sum of these different approximations can lead to errors. In addition, the numerical scheme of the model used is also a source of uncertainty itself, a modeling error of 5% for a coverage factor of k = 2 was found by [50].

### 3.3. Study Case: Karthala Hike (12 January 2020)

UVI measurements obtained with the Solarmeter collected during the ascension of the Karthala from Mvouni village on 12 January 2020 (called UVI-HIKE hereafter). UVI-HIKE measurements are represented on the first panel of Figure 6 (orange line) along with the corresponding UVI-RADIO-AS measured at Moroni on the same day (blue line), the corresponding modeled UVI-TUV-CS (green line) at the same altitude as UVI-HIKE and the altitude profile of the ascension (black line). On the second panel, the CF measured at Moroni is represented along with the environmental observations reported during the hike. On the last panel, a map of the area shows the locations of Moroni, Mvouni, and Mount Karthala, as well as the route taken during the hike (red line). An anticorrelation can be observed between UVI-RADIO-AS (at Moroni, Figure 6a) and the total CF measured at Moroni (Figure 6b). Cloudiness has a strong influence on the variability of the UVI on the ground. In the case of the ascent, during the morning UVI-HIKE increases until it reaches a UVI value of 20 at local noon. Then, after a sharp drop around 12:30 local time, a new UVI maximum of about 21 is reached around 13:00. The rest of the afternoon is characterized by very low UVI levels. The low UVI in the afternoon is related to the significant presence of clouds and even haze. The observed UVI maxima appear when broken clouds are present. Moreover, the UVI-HIKE maximum values even exceed the modeled UVI-TUV-CS at the same altitude: at 12:30 UVI-HIKE units is higher by 2.5 UVI units and, at 13:00, UVI-HIKE is higher by about 4. These maxima are probably due to the broken clouds and their ability to backscatter UV radiation, thus amplifying the UVI on the ground [16].

Standard erythemal doses (SEDs) and cumulative standard erythemal doses (CSEDs) for the day were also calculated in this study. SED and cumulative SED are shown in Figure 7. To compute SED, the erythemal irradiance is accumulated over a one-minute window. In the first panel (Figure 7a), the diurnal cycle of the SED is shown for the day of 12 January 2020, i.e., during the ascent of Mount Karthala. The SED calculated from UVI-RADIO, located in Moroni, is represented by a blue line, SED calculated from UVI-HIKE is an orange line. On the second panel (Figure 7b), the cumulative SED during the day is shown in blue for Moroni and in orange for the hike. The tolerance thresholds defined by the Fitzpatrick scale [52] are also shown for the two extreme phototypes (I and VI). In the last panel (Figure 7c), the cumulative SED over the day (C-SED) is represented for the whole study period calculated from UVI-RADIO and UVI-TUV-CS. The tolerance thresholds are also plotted on this figure. The SEDs observed in Moroni and on Mount Karthala are of the same order of magnitude with maxima reaching values of about 0.25 SED·min^−1^ at Moroni and about 0.3 SED·min^−1^ on Mount Karthala (Figure 7a). Very high SEDs are observed on Mount Karthala between 12:00 and 14:00 local time. These high values are related to the high UVI values observed previously (Figure 6a) and are thus probably due to the increase of UVI by the clouds. The cumulative SED for the day exceeds the Fitzpatrick threshold as early as 09:00 for phototype I at both Moroni and Mount Karthala (Figure 7b). For phototype VI, the threshold is exceeded from around 11:00 local time. The maximum cumulative dose observed on 12 January 2020 was 60 SED on Mount Karthala and 50 SED in Moroni. It was reached at about 15:00 local time in the first case and at about 16:00 local time in the second case. Observations of the daily cumulative SED evolution during the whole study period (Figure 7c), reveals that the thresholds for both phototypes are exceeded almost all year round. Indeed, the cumulative daily doses measured by the ground radiometer (blue crosses) are mostly between 20 SED and 80 SED. The cumulative daily doses modeled in clear sky are between 80 and 40 SED. For no day of the year 2020 did the cumulative doses measured by the radiometer exceed those modeled by TUV. The TUV clear-sky modeling previously showed an overestimation of UVI of about 11%, which explains the discrepancy observed between the two daily cumulative doses measured by the radiometer and those calculated from the TUV modeling. Nonetheless, the high values observed all year long clearly indicate health risks for the exposed population.

## 4. Discussion

The UV-Indian Network for Monitoring and Investigating UVR in the Indian Ocean deployed a UVR and cloud measurement station in Moroni, on the large Comoros island of Ngazidja. In addition, a measurement campaign was carried out on 12 January 2020 from the village of Mvouni to the summit of the Karthala volcano. These two instrumental phases added to the collection of local meteorological and satellite data, and UV modeling allowed the health risk related to UVR in Comoros to be analyzed and understood. The analysis of the meteorological data highlighted the annual cycle of precipitation and temperatures marking the dry season, during the austral winter, and the wet season, during the austral summer. The monthly UVI climatology estimated in Comoros by the OMI instrument shows a UVI in clear-sky conditions and at local solar noon varying between 7 and 16, and a UVI in all-sky conditions varying between 4 and 11 one hour on average after the local solar noon. The UVI maxima are observed during the wet season. Taking into account the scale defined by the WHO into account, the frequency analysis of the observed UVIs shows very frequent exposure of the population (for more than 90% of the year) to high, very high, or extreme UVIs. These extreme climatological values of UVI show the importance of monitoring UVI levels in this region and communicating to the general public about the dangers associated with this exposure. Similar high frequencies of very high UVI were also reported by [10] for points in the same region of the globe (Cape Town, South Africa and Saint–Denis, Reunion Island)

The comparison of UVI measurements from the radiometer (UVI-RADIO) installed in Moroni with satellite estimates (TROPOMI) or models (CAMS and TUV) shows a significant consistency between these datasets under clear sky conditions; compared to that of UVI-RADIO, correlation coefficients of 0.93, 0.98, and 0.95 were found for UVI-TROPOMI-CS, UVI-CAMS-CS and UVI-TU-CS. The correlation is lower under all-sky conditions, which is attributed to the poor ability of the models and satellites to characterize the cloud cover on the ground. This discrepancy is marked by a high standard deviation when the mean and relative differences are calculated in all-sky conditions. Moreover, a slight overestimation by the satellites and models is observed. Under clear-sky conditions, the relative differences are 5.7 ± 8.8%, 9.9 ± 11.9%, and 10.7 ± 12.7% for UVI-TROPOMI-CS, UVI-CAMS-CS, and UVI-TUV-CS-SUBSET, respectively. The discrepancy between ground measurements and satellite or model estimates may be partly due to radiometer drift following its recalibration in 2019. The comparison of TROPOMI measurements with other measurements was also studied by [41]. Relative median differences of the order of 10 to 20% were also found for four stations in the southwestern Indian Ocean region. CAMS outputs from the UVI were analyzed and compared with ground-based measurements made in Europe [53]. The observed differences were about ±0.5 UVI for more than 70% of the data. For UVI values of 10, this represents a relative difference of about 5% and for lower values this difference is about 10%. The UVI values in Europe are rather in the order of 3 to 10 UVI [54]. This gives an error of between 5 and 30%. This is the same order of magnitude obtained here in our study in the case of clear-sky measurements. Satellites and models have the advantage that they can provide global coverage. The comparison with ground measurements shows that satellites and models obtain close values in this region of the globe under clear sky conditions. However, this is not yet the case in cloudy conditions. It would therefore be necessary to continue the study of the impact of cloudiness on UVI to better estimate UVI using satellites or models.

Analysis of the UVI measurements made during the Karthala hike shows a record number of UVI maxima (about 20), which are attributed to the impact of the high altitude and fractional cloud cover. They highlight the greater health risk to the population living and working at medium and high altitudes. The maximum SEDs calculated during this campaign were about 0.30 SED and 0.24 SED over Karthala and Moroni, respectively. Cumulated over the day, we found that the health risk thresholds were exceeded from 09:00 in the morning for phototype I skin and from 11:00 for phototype VI skin, and, over the whole day, the maximum accumulation was 60 SED on Karthala and 50 SED at Moroni. These SED and cumulative SED values (over 50 SED) are close to those obtained in this region of the globe at similar latitudes by [13]. In the northern hemisphere, in Spain, cumulative SED values of over 60 SED were also reported by [55]. The cumulative SEDs over the day for the entire study period presented here show that the health risk thresholds were exceeded in Moroni throughout the year and for any skin type. The high level of cumulative doses throughout the year reinforces the need to monitor UVI in this region and to communicate preventive messages to the general public about the risk of UVR. Few studies exist today to quantify the mortality rate related to this risk, so it is important to study this risk in more detail to understand and prevent it in the future.

## 5. Conclusions

This study shows that the health hazards related to ultraviolet radiation (UVR) in the Comoros are very high. According to the WHO scale, UVI levels measured at solar noon are classified as extreme throughout the year for more than 50% of cases, and more than high for 90% of cases throughout the year. In addition, the daily doses of UV received largely exceed the health thresholds throughout the year for all skin phototypes. Although the risk associated with these parameters depends on the exposure of the populations. The risk is significant; indeed, exposure is high due to the large proportion of the population living in rural areas and the fact that the country is predominantly agricultural. The vulnerability of the population of the Union of the Comoros to the health risks of UV radiation (skin cancer, cataracts, etc.) is therefore significant.

The results showed that the studied model and satellite data were in good agreement with surface UVI measurements under clear sky conditions. However, this was not the case in cloudy conditions and further studies are needed to analyse the reasons. In general, satellite and model data showed overestimation compared to ground-based UVI measurements, but during measurements at Mount Karthala and some other specific conditions they showed underestimation. This suggests increase in the surface UVI due to cloud cover.

## Figures and Tables

**Figure 1 ijerph-18-10475-f001:**
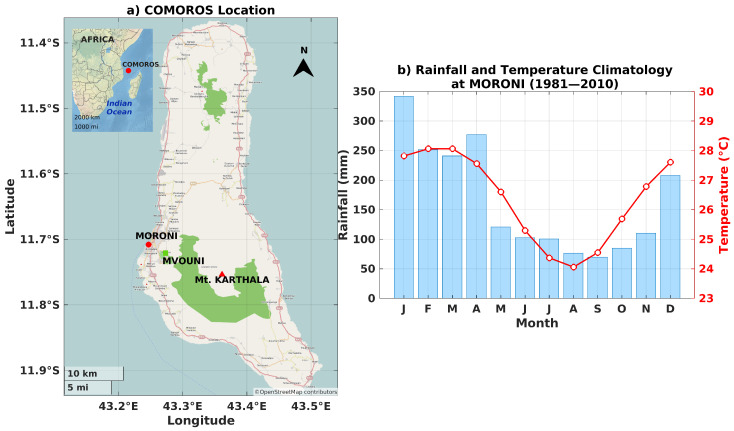
Context ( Map and Precipitation Climatology). (**a**) Comoros Location: town of Moroni (red dot), Mvouni Village (green square), and Mount Karthala (red triangle). (**b**) Rainfall (barplot) and temperature (red line) climatology at Moroni from 1981 to 2010 period.

**Figure 2 ijerph-18-10475-f002:**
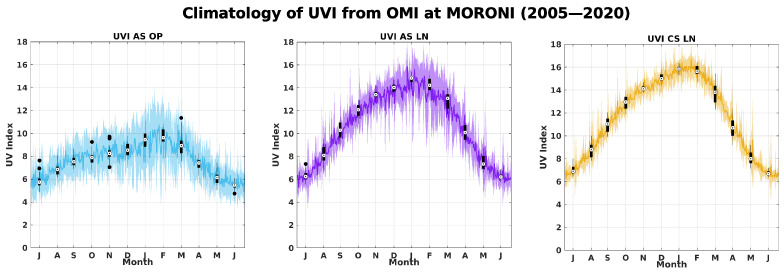
Monthly Climatology of UVI from OMI. Black box diagrams for each month are centered on 15th day of month. Median is central value, box edges are set at 25 th and 75 th percentiles, whiskers show extreme values, excluding outliers, which are represented by single black circles. Blue, purple, and yellow lines show mean for each day of year, and blue and yellow shading shows one standard deviation. Months on X-axis were reorganized to highlight austral summer. Blue color denotes UVI-OMI-AS, and yellow color denotes UVI-OMI-CS.

**Figure 3 ijerph-18-10475-f003:**
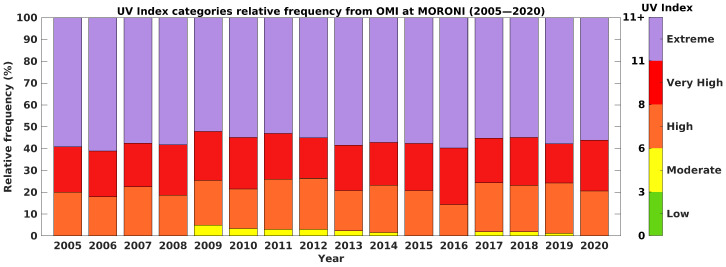
Annual relative frequency of UVI thresholds from UVI-OMI-AS-LN during 2005–2020 period. Colorbar represents UVI-OMI-AS-LN (UVI all-sky at local noon), according to color scale defined by OMS: Low UVI from 0 to 3 in green; moderate UVI from 3 to 6 in yellow; high UVI from 6 to 8 in orange; very high UVI from 8 to 11 in red, and extreme UVI above 11 in purple.

**Figure 4 ijerph-18-10475-f004:**
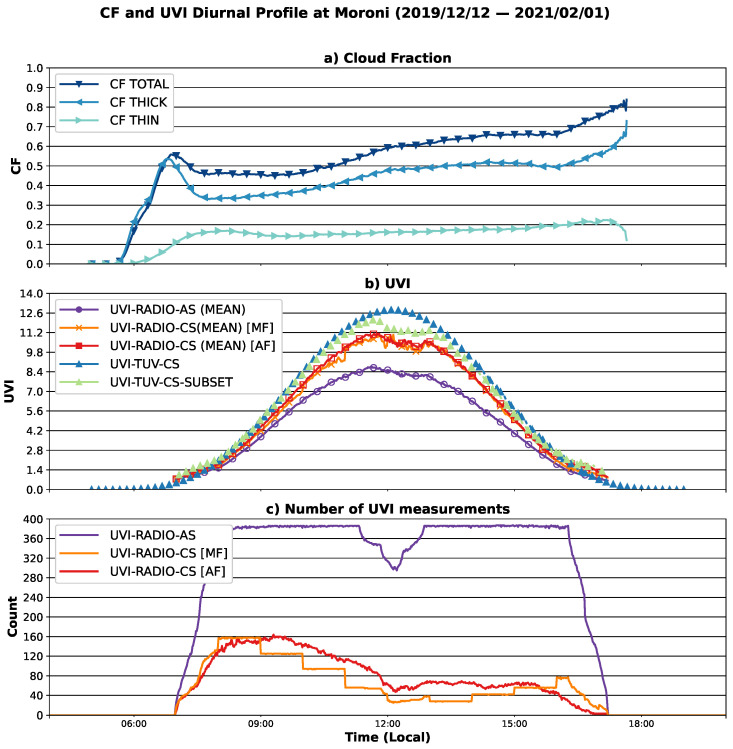
Mean diurnal cycle of cloud fraction and UVI. (**a**) Total, thick, and thin cloud fraction. (**b**) Mean UVI in all sky (purple circle). (**c**) Number of UVI measurements.

**Figure 5 ijerph-18-10475-f005:**
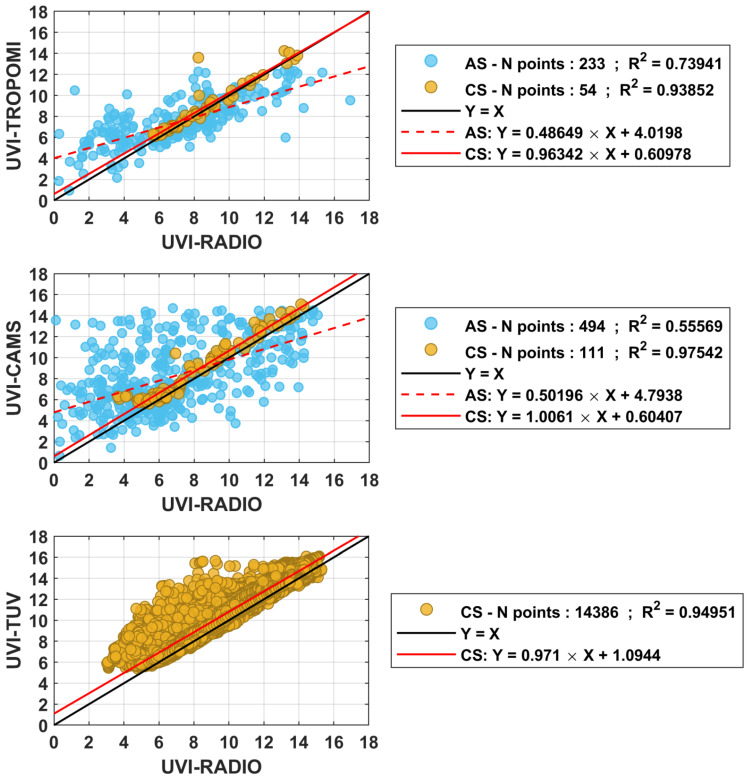
Correlation between UVI Satellite estimates or UVI Model results and UVI-Radio. All-Sky UVI are in blue circles. Linear fitting is represented by a red dashed line. Clear sky UVI are in yellow circles. Linear fitting is in solid red line.

**Figure 6 ijerph-18-10475-f006:**
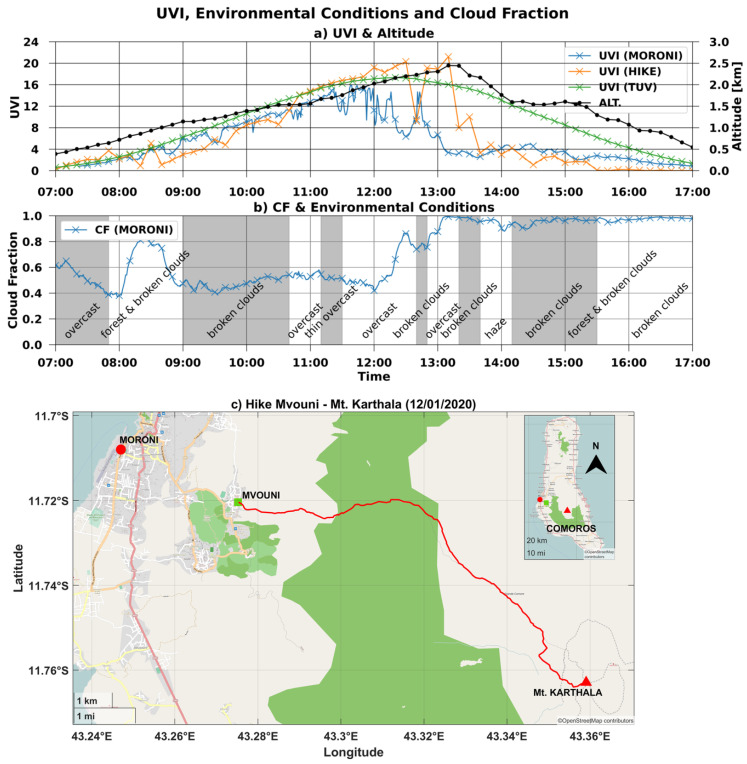
(**a**) UVI, (**b**) cloud fraction, and environmental observations during the hike of 12 January 2020. (**c**) Map showing location of town of Moroni, village of Mvouni, and summit of Karthala, with route taken for hike.

**Figure 7 ijerph-18-10475-f007:**
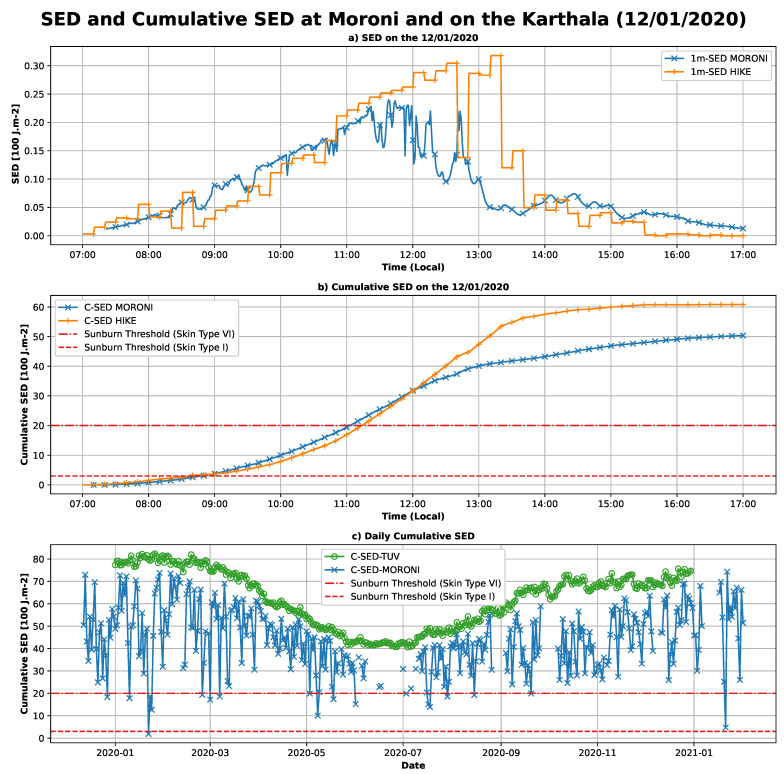
(**a**) Equivalent erythemal doses (erythemal radiant exposure of 100 SED) measured at Moroni and during ascent of Karthala on 12 January 2020. (**b**) Cumulative doses at Moroni (orange) and during ascent of Karthala (blue) on 12 January 2020. Thresholds for phototypes of two erythemal extremes are in red dashed and dash dotted lines (according to Fitzpatrick). (**c**) Daily cumulative erythemal doses: in blue at Moroni (RADIO), and in green at Moroni (TUV). Thresholds for phototypes of two extremes erythema are represented in red (according to Fitzpatrick).

**Table 1 ijerph-18-10475-t001:** Ground-based and satellite measurements of UVI.

Dataset	Platform	Type	Resolution	Field Used for Computation/Calibration		Temporal Coverage	References
				Ozone Field Used	Aerosol Field Used		
RADIOMETER	Ground-Based	Radiometer Kipp&Zonen UVS-E-T	dt = 1 or 5 min	OMUVG	None	December 2019–March 2021	Cadet et al. [12]
OMUVBG	Satellite	Spectrometer Measurement at 360 nm then LUT	Daily Overpass/Solar Noon 0.25 × 0.25 deg	OMI	Krotkov et al. [38] Herman et al. [39]	2004–2020	Tanskanen et al. [40] Arola et al. [34] Levelt et al. [32]
TROPOMI	Satellite	Spectrometer Measurement at 354 nm then LUT	Daily Overpass/Solar Noon 5.6 × 3.7 km	TROPOMI L2 total ozone column product Garane et al. [37]	Aerosol Climatology Kinne et al. [35]	November 2017–April 2021	Lindfors et al. [36] Lakkala et al. [41]
CAMS	Model	Modelling	dt = 6 h 40 km (after 21 June 2016) 80 km (before 21 June 2016)	Modelled	Modelled	June 2016–April 2021	Bozzo et al. [42]

**Table 2 ijerph-18-10475-t002:** Comparison results between UVI-TROPOMI or UVI-CAMS or UVI-TUV against UVI-RADIO in both all-sky and clear-sky conditions. RD stands for Relative Difference and AD for absolute difference. ±1 Standard deviation is also provided for these difference. R is correlation coefficient, and Ndata is number of data points compared.

Stats	UVI-TROPOMI-AS	UVI-TROPOMI-CS	UVI-CAMS-AS	UVI-CAMS-CS	UVI-TUV-CS-SUBSET
Mean RD [%]	63.4 ± 183.1	5.8 ± 8.8	93.1 ± 552.4	9.9 ± 11.9	10.7 ± 12.7
Mean AD [UVI]	2.0 ± 1.7	0.5 ± 0.7	2.4 ± 2.4	0.7 ± 0.6	0.9 ± 0.7
Median RD [%]	19.7	3.4	23.3	7.1	7.5
Median AD [UVI]	1.6	0.3	1.7	0.4	0.7
R^2^	0.68	0.93	0.55	0.98	0.95
Ndata	416	62	577	111	14,386

## Data Availability

Data from the UV radiometers and the total sky imagers can be downloaded from Zenodo and are referenced under the following doi: https://doi.org/10.5281/zenodo.4811488, accessed on 1 July 2021 [56]. The OMUVBG data can be downloaded through the website at https://disc.gsfc.nasa.gov/datasets/OMUVBG_003, accessed on 1 July 2021. The overpass files for the TROPOMI product version 1.02.02 (overpass extractor version 1.02.00) were downloaded from the FTP site accessible via the product web page at https://nsdc.fmi.fi/data/data_s5puv.php, accessed on 1 July 2021. The CAMS data can be downloaded through the website at https://apps.ecmwf.int, accessed on 1 July 2021.

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
