# Peer review of "Monitoring Solar Radiation UV Exposure in the Comoros"

_ijerph, 2021, doi:10.3390/ijerph181910475_

Round 1
Reviewer 1 Report
Article IJERPH-13382130 Revision: 16 September 2021
Monitoring Solar Radiation UV Exposure in the Comoros.
Kevin Lamy 1 https://orcid.org/0000-0002-9115-1319*, Marion Ranaivombola 1, Hassan Bencherif 1,2, Thierry
Portafaix 1, Mohamed Abdoulwahab Toihir 5, Kaisa Lakkala 3,4, Antti Arola 4, Jukka Kujanpää 3, Mikko R.A.
Pitkänen 3,4, Jean-Maurice Cadet
Abstract: As part of the UV-Indien project, a station for measuring ultraviolet radiation and the cloud fraction was installed in December 2019 in Moroni, the capital of the Comoros, situated on the west coast of the island of Ngazidja. A ground measurement campaign was also carried out on 12th January 2020 during the ascent of Mount Karthala, located in the centre of the island of Ngazidja. In addition, satellite estimates (Ozone Monitoring Instrument and TROPOspheric Monitoring Instrument) and model outputs (Copernicus Atmospheric Monitoring Service and Tropospheric Ultraviolet Model) have been combined for this same region. On the one hand, these different measurements and estimates make it possible to quantify, evaluate and monitor the health risk linked to exposure to ultraviolet radiation in this region and, on the other, they help to understand how cloud cover influences the variability of UV radiation on the ground. The measurements of the Ozone Monitoring Instrument on board the EOS-AURA satellite, being the longest time series of ultraviolet measurements available in this region, make it possible to quantify the meteorological conditions in Moroni and to show that more than 80% of the ultraviolet indices are classified as high, and that 60% of these are classified as extreme. The cloud cover measured in Moroni by an All Sky Camera was used to distinguish between the cases of ultraviolet index measurements taken under clear or cloudy sky conditions. The ground-based measurements thus made it possible to describe the variability of the diurnal cycle of the ultraviolet index and the influence of cloud cover on this parameter. They also permitted the satellite measurements and the results of the simulations to be validated. In clear sky conditions, a relative difference of 20 between 6 and 11% was obtained between satellite or model estimates and ground measurements. The ultraviolet index measurement campaign on Mount Karthala showed maximum o ne-minute standard erythemal doses at 0.3 J.m2 and very high daily cumulative erythemal doses, at more than 80 J.m2. These very high levels are also observed throughout the year and all skin phototypes can exceed the daily erythemal dose threshold, at more than 20 J.m2.
Keywords: ultraviolet radiation; clouds; tropics; observations, modelling, erythemal doses
Reviewer Recommendations:
1 Normally when high number of data series are used, a technique, which is called “data quality control” should be applied on each data series. Excuse me but I do not have find along the text, this study. Perhaps the manuscript have used other name.
2-Line 367:- When the symbols +and – are used, they should be written with a format similar to the WORD text processing does. Please verified all of these symbols used along the manuscript.
3- About the Table 2. The values and errors. There are some specific criteria for writing errors. In this case, errors have very big number of figures and as consequence it would be well written. Please, errors should have a number of figures and should in the Figures,if be written following a specific methodology. For example sometimes it is necessary to eliminate some decimal figures but taken into account its values. This theory is explained in specific text, about “Measurements and errors”. I recommend to see it.
4-. In addition, error values sometimes are shown into the Figures when authors consider that manuscript readers increase the knowledge and the results are more clear.
5- Figure 6:- It shows two times the figure b) ,perhaps is better : a, b and c .????
6- Figure 7: It is composed by a, b and c figures as the legends show. But it would be better if a, b and c are also indicated on each of the figure.
7-Line 465: some units should be written as exponential. example: Jm-2, it is Jm-2
This error is in more than one place. Please verified.
Conclusions: The manuscript has a lot work, it is well developed and the results are very important. The study is very interesting due to the effects of UV on people in this region. In places with UVI of 10, sun radiation is dangerous, in places with UVI till 20 it will be more dangerous.
Reviewer 2 Report
This study combined ground-observed, satellite-retrieved, and modeled UVI data to understand the diurnal variability of UV radiation, the influence from cloud cover, and the linked health risk. The work is meaningful, but I still have some questions about their analysis applied and their main conclusions.
- Line 101,the full form of TOC should be provided here.
- Line 107, since the field measurements start on 12 January 2020, so the ground observations applied in this research are only of 1 year?
- The form of citations are not consistent over the manuscript, exceptions are e.g. Lines 127, 137-138
- Line 140, replace “are” by “were”
- Line 164, but you still use “UV index” in the rest of your study. How to understand your UVI index or UVI-TROPOMI? 10 means high or low level? Such information should be clearly provided in your Method section.
- Table 1, a large difference can be noticed in the spatial resolution of the applied data products. How can satellite and model data, showing such large pixel size, effectively represent the situation at station-scale? Please also include the information of the temporal coverage of the different datasets applied in your research.
- Line 254 and Figure 2, need to explain what are UVI-OMI-AS-OP, UVI-AS-LN and UVI-OMI-CS-LN in the text or in the caption of Figure 2.
- Lines 256-263, can be deleted, duplicate content from the caption of Figure 2
- Lines 263-265, the difference is not clear from Figure 2, can you explain why there is no difference in UV index between all-sky and clear-sky conditions?
- Figure 3, what is the data source for Figure 3? Since your field measurements start in 2020, were did you get the records for 2005-2020?
- Lines 282-290, not necessary to show in the main text, but better to put in the caption of Figure 4
- Figure 4, I don’t understand what are UVI-RADIO-CS-MF, UVI-TUV-CS, UVI-RADIO-AS, etc. and their differences.
- From your Figure 4, I cannot tell a clear correspondence between the diurnal cycles of CF and UVI. I didn’t see “their maxima are amplified by the presence of fractionated clouds” as concluded by the authors.
- Lines 324-347, better to be addressed in the caption of Figure 5
- Table 2, what are the values after plus and minus signs?
- Line 377, I don’t think using aerosol climatology can cause a bias in the climatological means between ground and modelled data.
- Lines 382-388, not needed
- Figure 6, why UVI wasn’t amplified by the high level of broken clouds at 08:00-09:00?
- Line 416, now clouds increase UVI?
- Discussion section, surprised to see zero citation at a Discussion section, can move to the Conclusion section
- Conclusion section, seems more like a Discussion section, though also with zero citation
Reviewer 3 Report
General discussion
The manuscript is well written. The methodology is adequate from a meteorological point of view. The authors make correct use of the information available. From the technical point of view, I only have to suggest minor changes.
However, I think the authors should discuss more in depth the scientific interest of this paper. Everyone knows that radiation exposure in tropical latitudes is high. What conclusions can we draw from these results? Do the authors think there could be an increase in UV radiation related to climate change? Is the UV exposure higher in the Comoros than in other similar latitudes? Do they think the health authorities should take any action? I would appreciate some discussion on these points.
Minor changes
- Line 101: Define TOC at the first appearance.
- Line 100-101: “These indices are largely due to the country’s tropical location, where the incoming solar radiation is most intense and the TOC is the lowest.”
This is not exactly correct. TOC is lower over tropical latitudes tan over mid latitudes. However, the lowest TOC is located over polar regions.
- Figure 2 caption. When they refer to “the blue and yellow lines” I guess they mean “the blue, purple and yellow lines”.
- Line 326. It is not correctly written. Maybe “UVI-TUV-CS-SUBSET” should be removed.
